# Behavior-change interventions to improve antimicrobial stewardship in human health, animal health, and livestock agriculture: A systematic review

**Jessica Craig**[1]*, **Aditi Sriram**[2], **Rachel Sadoff**[3], **Sarah Bennett**[4], **Felix Bahati**[4,5,6], **Wendy Beauvais**[1]

**1** Department of Comparative Pathobiology, College of Veterinary Medicine, Purdue University, West Lafayette, Indiana, United States of America, **2** One Health Trust, Bangalore, India, **3** Mailman School of Public Health, Columbia University, New York, NY, United States of America, **4** King's College London, London, United Kingdom, **5** KEMRI Wellcome Trust Research Programme, Health Services Research Unit, Nairobi, Kenya, **6** Department of Environmental Health and Disease Control, College of Health Sciences, Jomo Kenyatta University of Agriculture and Technology, Nairobi, Kenya

* craig119@purdue.edu

**Data Availability Statement:** All data is provided in the supplementary materials.

## Abstract

Antimicrobial resistance (AMR) is an economic, food security, and global health threat accelerated by a multitude of factors including the overuse and misuse of antimicrobials in the human health, animal health, and agriculture sectors. Given the rapid emergence and spread of AMR and the relative lack of development of new antimicrobials or alternative therapies, there is a need to develop and implement non-pharmaceutical AMR mitigation policies and interventions that improve antimicrobial stewardship (AMS) practices across all sectors where antimicrobials are used. We conducted a systematic literature review per the Preferred Reporting Items for Systematic Reviews and Meta-Analyses guidelines to identify peer-reviewed studies that described behavior-change interventions that aimed to improve AMS and/or reduce inappropriate antimicrobial use (AMU) among human health, animal health, and livestock agriculture stakeholders. We identified 301 total publications– 11 in the animal health sector and 290 in the human health sector–and assessed described interventions using metrics across five thematic areas- (1) AMU, (2) adherence to clinical guidelines, (3) AMS, (4) AMR, and (5) clinical outcomes. The lack of studies describing the animal health sector precluded a meta-analysis. Variation across intervention type, study type, and outcome precluded a meta-analysis for studies describing the human health sector; however, a summary descriptive analysis was conducted. Among studies in the human health sector, 35.7% reported significant ($p<0.05$) pre- to post-intervention decreases in AMU, 73.7% reported significant improvements in adherence of antimicrobial therapies to clinical guidelines, 45% demonstrated significant improvements in AMS practices, 45.5% reported significant decreases in the proportion of isolates that were resistant to antibiotics or the proportion of patients with drug-resistant infections across 17 antimicrobial-organism combinations. Few studies reported significant changes in clinical outcomes. We did not identify any

**Funding:** This project was funded by Purdue University, the One Health Trust, and Tephinet.

**Competing interests:** The authors have declared that no competing interests exist.

overarching intervention type nor characteristics associated with successful improvement in AMS, AMR, AMU, adherence, nor clinical outcomes.

## Introduction

Antimicrobial resistance (AMR) is an economic, food security, and global health threat [1–3]. Globally, there were an estimated 4.95 million human deaths associated with drug-resistant bacterial infections in 2019 [4]. The Centers for Disease Control and Prevention estimates that the annual cost to treat hospital- and community-acquired drug-resistant bacterial infections in the US alone is $4.6 billion due to prolonged hospital stays, more complex healthcare needs, and treatment with second- and third-line antimicrobial therapies [5]. In addition to the human health and economic burden, AMR among pathogens affecting animals has adverse animal welfare impacts; imposes additional treatment costs on animal owners thereby increasing food production costs; and may impact international food and animal trade [3, 6]. Researchers estimate that AMR may cause a 7.5 percent decline in global livestock production by 2050 [7].

Drug resistance is accelerated by a multitude of factors including the overuse and misuse of antimicrobials in the human health, animal health, and agriculture sectors. Although antimicrobial use (AMU) and consumption (AMC) surveillance remains limited in the human and animal health sectors, available evidence demonstrates that AMU is increasing globally. Analysis of AMC rates, approximated from data on national-level antimicrobial sales data, indicated a 39% increase in per capita AMC in the human health sector between 2000 and 2015, rising from 11.3 to 15.7 daily defined doses per 1,000 people per day (DIDs) [8]. Concerningly, the use of World Health Organization-designated "Watch" antibiotics has increased faster than "Access" antibiotics. Between 2000 and 2015, there was a 90.9% increase in the global per capita consumption of "Watch" antibiotics, from 3.3 to 6.3, compared to a 26.2% increase in global per capita consumption of "Access" antibiotics which rose from 8.4 to 10.6 DIDs in the same period [9]. Previous studies show clinically inappropriate AMU rates as high as 55% in South Africa, 88% in Pakistan, and 61% in China in the human health sector; however, emerging evidence suggests there is significant heterogeneity in AMU practices across countries and clinical settings [10–13]. Best available evidence suggests that AMU in the animal health sector is rising as the global demand for animal products increases [14]. In 2017, there were an estimated 93,309 tons (95% confidence interval [CI]: 64,443–149,886) of antimicrobials sold for use in chicken, cattle, and pig systems across 41 countries. In the same year, AMC in the aquaculture sector was estimated at 10,259 tons across six fish species in 33 countries [15]. By 2030, AMU is projected to rise to 104,079 tons among chicken, cattle, and pigs and to 13,600 tons among fish [14, 15].

Concerningly, the emergence and evolution of drug-resistant microorganisms is outpacing the development of new antimicrobial agents. In the past 50 years, no new classes of antimicrobial agents active against Gram-negative bacteria have been brought to market, and only 5 of the 20 major global pharmaceutical companies are engaged in antimicrobial research and development (R&D) [16]. Investment in antimicrobial R&D is not considered lucrative given the challenges in antimicrobial discovery, rapid introduction of generic formulations, and the speed at which microorganisms develop resistance to antimicrobials. Moreover, new antimicrobials are likely to be reserved for special or emergency use to preserve their efficacy; therefore, there would paradoxically be less demand for these drugs.

While AMR poses an increasing public health threat, the number of deaths that could be prevented by improving access to antimicrobials, about 5.7 million in low- and middle-income

countries (LMICs), exceeds the morbidity and mortality burden from AMR indicating that access to clinically appropriate antimicrobials for the treatment of infectious diseases remains a critical issue [16].

Given the health and economic impacts of AMR, challenges around access to appropriate antimicrobial therapies, and the lack of antimicrobial R&D, there is an urgent need to develop and implement impactful non-pharmaceutical AMR mitigation policies and interventions that improve antimicrobial stewardship (AMS) practices in the human health, animal health, and agriculture sectors. There is a growing body of literature describing such non-pharmaceutical AMS interventions which include education, training, and health information campaigns to improve awareness and knowledge among technical stakeholders and the public; installing AMS committees in healthcare facilities to approve and/or review prescription practices; and developing and implementing standard clinical treatment guidelines or clinical treatment algorithms which provide clinicians with an evidence-based resource to guide clinical decision making around AMU and prescription [17–19]. Despite this emerging evidence, there remains no consensus on what constitutes the most impactful or cost-effective AMS practices across various sectors and settings. Therefore, the purpose of this study was to conduct a systematic literature review and meta-analysis to identify behavior-change interventions aimed at improving AMS and AMU across the human health, animal health, and agriculture sectors to identify gaps in the evidence base or to describe trends towards best AMS practices in various resource and income settings.

## Materials and methods

This study followed the Preferred Reporting Items for Systematic Reviews and Meta-Analyses (PRISMA) guidelines [20]. We searched PubMed, Web of Science, Embase, Centre for Agriculture and Bioscience International (CABI), and the Cochrane Database of Systematic Reviews for peer-reviewed studies that described behavior-change interventions that aimed to improve AMS and/or reduce inappropriate AMU or AMC among human health, animal health, and livestock agriculture stakeholders including but not limited to health providers, patients, farmers, or animal owners [21–25]. Two searches were conducted; the first on 15 June 2021 identified studies published prior to 15 June 2021 and a second search conducted on 31 August 2022 identified studies published between 15 June 2021 and 31 August 2022 (inclusive). The following search string was used to search the title and abstracts of published entries within each database: (intervention OR trial) AND (antimicrobial OR antibiotic) AND (use OR stewardship OR consumption OR prescription).

Two researchers independently screened returned entries for eligibility in a two-step process: a review of the title and abstract followed by a full-text review, according to the following criteria: (1) Only scientific articles published in peer-reviewed journals were included; pre-prints, course completion papers or theses, and internal reports or policy briefs were excluded. (2) Papers published in any language and written on interventions conducted in any country were included. (3) All study types including randomized controlled trials, pre/post study designs, and observational studies were included. Review articles were screened for additional relevant and non-duplicative studies. (4) Studies were included if they reported on at least one primary outcome which fell into three general categories: (a) AMU or AMC; (b) adherence/compliance of antimicrobial therapy to facility, national, or international clinical treatment guidelines; and (c) rates of specific stewardship practices such as the intravenous to oral therapy conversion rates, frequency of obtaining specimen cultures for pathogen identification and/or antimicrobial susceptibility tests. For studies that reported on at least one primary outcome, we also extracted data for secondary outcomes which included (a) the incidence or

prevalence of drug-resistant infections; (b) length of stay in the healthcare facility or specific unit (such as intensive care); (c) health outcomes such as patient/animal mortality or rehospitalization rates; and (d) the economic or cost impact of interventions. (5) Non-relevant papers that did not describe interventions aimed at changing behaviors to optimize AMU, AMC, or AMS in the human health, animal health, or livestock agriculture sectors or those that did not report on our primary outcomes of interest were excluded. Therefore, studies such as those that reported solely on changes in knowledge, attitudes, and perceptions regarding AMR or AMU were excluded as were studies that assessed drug or drug regimen efficacy. (6) Studies were also excluded if the full text was unavailable or unobtainable, if results and statistics were not presented numerically (such as only presented graphically), or if insufficient data was reported to conduct statistical analyses such as to calculate odds ratios, risk ratios, or rate ratios or to conduct t-tests or Chi-square tests. At each stage, two reviewers independently screened entries for eligibility, identified discrepancies, and re-evaluated articles until a consensus was reached. Duplicates were identified when results from each search were combined into a single database.

Data extraction was conducted by one researcher (JC, SB, FB) and reviewed by a second (JC, SB, and FB); data was compiled using Microsoft Excel software. We used a standardized spreadsheet (S4 File) to extract data from each publication that met our inclusion criteria, including the publication year, country and setting on the intervention, type of evaluation, type of study, description of intervention, intervention start and end date and duration, total sample size and unit of sample size, the study's outcome, and all relevant numerical data reported. The income classification of the country where the intervention occurred, per the World Bank's 2021 classification, was also recorded [26]. We classified studies as being either single or multi-center and as either single or bundled (meaning more than one type of intervention occurred), based on the description of the intervention. Duration of intervention was computed, if not provided in the study, using the intervention start and end dates.

For the meta-analysis, we attempted to create syntheses based on three criteria: type of statistic reported, study type, and outcome variables. First, studies were categorized by type of statistic (rates or proportions) reported. Then studies were categorized by study type or data analysis/reporting methodology. These categories included: pre/post observational studies; randomized or non-randomized control trials (RCTs) that compared a control arm to an experimental/intervention arm; randomized or non-randomized control trials that compared one intervention arm to another intervention arm; and time series studies. We then categorized the studies by outcome. We aimed to conduct meta-analyses for all syntheses that had at least 3 studies. To compare the study arms or pre- to post-intervention changes according to the study outcomes, we calculated odds ratios, rate ratios, or risk ratios depending on the type of outcome data reported; we also calculated the. 95% CI, chi-square statistic, Wilcoxon signed-rank test, and independent or dependent t-test statistics were calculated, as appropriate [27]. Bivariate analyses and linear and logistic regression analyses were conducted across studies to identify trends in intervention success in various settings. For logistic regression analyses for the pre- to post-intervention studies, a categorical variable was created with 3 categories indicating if there was a significant increase, significant decrease, or no statistically significant change in pre- to post-intervention outcomes. Backwards stepwise elimination process based on likelihood ratio tests was used to build final regression models. We utilized funnel plots and the Egger's regression test to assess for publication bias; standard error were calculated from confidence intervals as described previously by Cochrane [28]. We used the Quality Assessment Tool for Quantitative Studies to assess study quality and risk of bias [29]. For the quality assessment, each study was independently reviewed by two researchers with a third to resolve disputes and compute a consensus quality score.

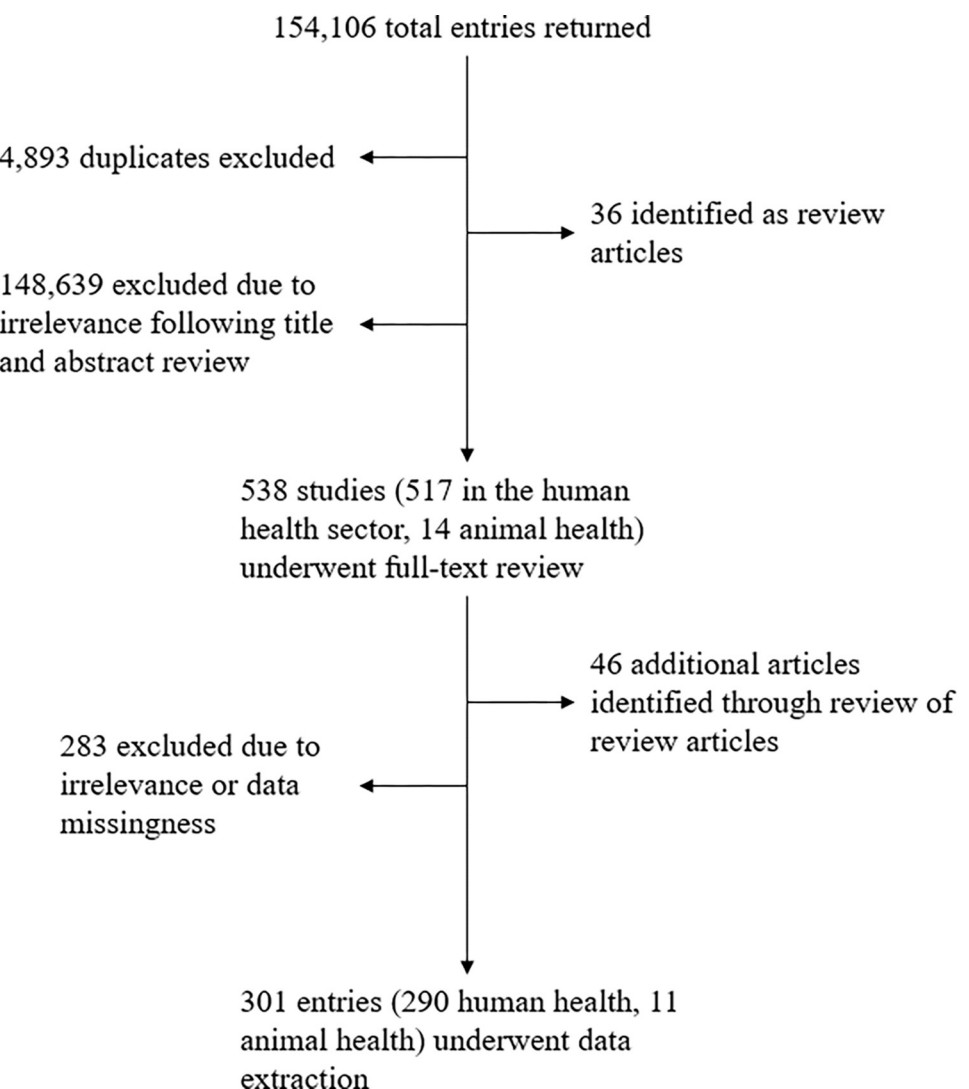

**Fig 1. Flow diagram of systematic literature review describing behavior change interventions to improve antimicrobial use and stewardship in the animal and human health sectors.**

## Results

Our review returned a total of 154,106 publications of which 4,893 were duplicates; 148,639 were determined to be irrelevant based on the title and abstract review; and 36 were identified as review articles. Following full-text review, a total of 301 publications– 11 in the animal health sector and 290 in the human health sector (S1 and S2 Files)–met the study's inclusion criteria and underwent data extraction (Fig 1). Per the Quality Assessment Tool for Quantitative Studies, 4 of 11 studies that described interventions in the animal health or agriculture sectors (36.4%) were determined to have strong overall quality ratings, 3 (27.3%) had moderate quality ratings, and 4 (36.4%) had weak quality ratings while 129 of the 290 studies (44.5%) that described interventions in the human health sector were determined to have strong overall quality, 115 (39.7%) had moderate overall quality, and 46 (15.9%) had low overall quality scores (S3 File). A common reason for low quality scores was a high risk of selection bias or failing to control for potential cofounders. Funnel plots

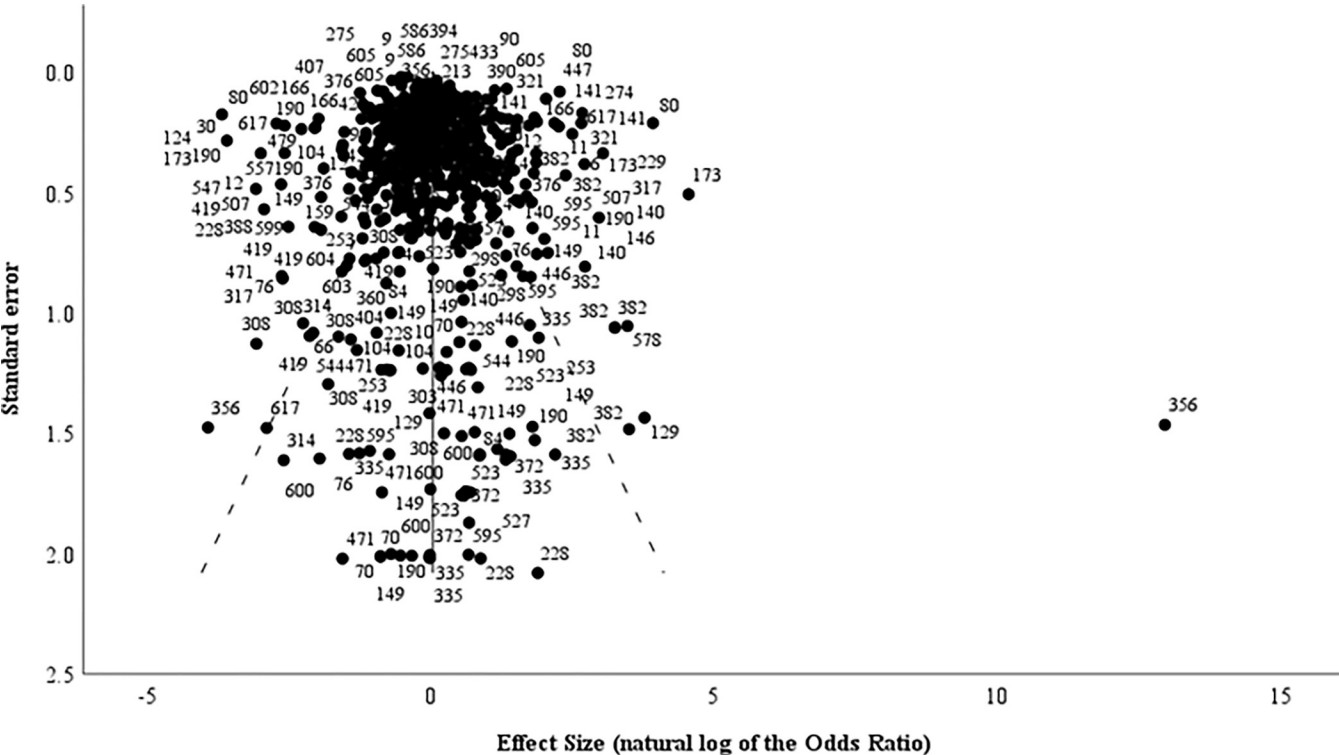

**Fig 2. Funnel plot demonstrating lack of publication bias across all studies that met our review's inclusion criteria.** Labels indicate study identification number. Egger's regression analysis indicated there was no significant publication bias (Coefficient: 0.169; p-value = 0.216). Funnel plot and Egger's regression analyses, including the calculation of standard error, followed previously described methodologies [28].

and Eggers regression analyses (p = 0.216) did not reveal statistically significant publication bias (Fig 2).

## Animal health and agriculture sectors

Only 11 studies conducted in the animal health or agriculture sector met our study's inclusion criteria. Given the small sample size and variation across the study types, outcomes, and types of statistics reported, a meta-analysis was not possible. Seven of the studies (63.6%) assessed the impact of interventions among food-producing animals including swine (n = 3), poultry (n = 1), dairy cows (n = 2), and calves (n = 1) [30–36]. Four studies (36.4%) assessed interventions in companion animals [37–40]. Year of study publication ranged from 2014 to 2021. Across the 11 studies, 8 countries were represented (Germany, Sweden, Netherlands, Belgium, France, Switzerland, the United Kingdom, and Vietnam). Interventions ranged from 5 months in duration to 5 years. Five studies (45.5%) implemented bundled interventions while the remaining 7 studies (63.6%) tested single interventions. Four studies (36.4%) implemented online AMS tools that either provided guidance on AMU and AMS or restricted antimicrobial prescriptions. Other interventions included veterinary guidance on husbandry, biosecurity, or AMU; educational programs; review and feedback of AMU; and programs to implement infection prevention and control measures. Overall AMU decreased significantly (p value < 0.05) for all but one study which implemented an online stewardship tool among calf farmers in Switzerland [31]. Three studies assessed pre- to post-intervention changes in animal health status and productivity and documented either no change or an improvement in mortality, animal weight gain, or feed conversion ratio associated with the intervention [30, 34, 35]. One

study that implemented a voluntary AMS program among pig farmers in France documented a 90% reduction in cephalosporin usage between 2010 and 2016 and demonstrated a reduction in drug-resistance among commensal and pathogenic *Escherichia coli* isolates sampled during the same period [33].

## Human health sector

Of the 290 studies that assessed behavior-change interventions in the human health sector, 40 (13.7%) were RCTs, 237 (81.7%) were pre/post observational studies, and the remaining 13 (4.5%) followed alternative study designs such as historical cohort study. Ninety-seven studies (33.4%) were retrospective, 129 (44.5%) were prospective, 49 studies (16.9%) utilized both retrospective and prospective evaluations, and the remaining studies did not specify the evaluation type utilized to assess the intervention. Year of publication ranged from 2001 (n = 2) to 2021 (n = 44). Studies were most commonly conducted in high-income countries (n = 228 studies, 78.6%) with 32, 21, and 3 studies (11.0, 7.2, 1.0%, respectively) conducted in upper-middle, lower-middle, and low-income countries, respectively. A total of 49 countries were represented across the studies that met our inclusion criteria; USA (n = 110 studies), Canada (n = 16), Japan (n = 15), Spain (n = 15), China (n = 12), and Italy (n = 10) were the most represented. Academic/teaching hospitals or tertiary care facilities were the most common setting for intervention implementation.

Given variation across the type of statistic reported, the study design, and the outcome variables reported by each study, meta-analysis syntheses contained fewer than our minimum of 3 studies. Therefore, our results will focus on a descriptive summary analysis of the studies by outcome theme.

**Theme 1: Antimicrobial use.** Of 290 studies that described interventions in the human healthcare setting and met our eligibility criteria, we extracted AMU data, quantified as a proportion of *patients* who received an antimicrobial before and after the study intervention, from 42 studies (Fig 3). The denominator (i.e. *patients*) was defined differently in each study e.g. patients presenting at a primary healthcare facility, patients in a long-term care facility. Amongst the 15 of 42 studies (35.7%) that found a statistically significant (p value <0.05) pre- to post-intervention decrease in AMU; the average decrease in the proportion of patients receiving various antimicrobial agents or classes was 10.4% (Standard deviation [SD]: 16.6) and the range was 3.1 to 58%. Four of these studies described interventions in lower-middle-income countries (Bangladesh, India, n = 2, Iran), two in upper-middle-income countries (China, Malaysia), and 9 in high-income countries (Finland, Hong Kong, Netherlands, Spain, UK, USA, n = 4). The clinical settings involved in these interventions included a veteran's hospital, a university dentistry practice, a long-term care facility, a pediatric hospital, primary healthcare or community clinics (n = 4 studies), secondary or tertiary hospitals (n = 5 studies). Across these 15 studies, 7 (46.7%) assessed single interventions and the remaining 8 (53.3%) tested bundled interventions; intervention duration ranged from 6 weeks to 3 years. Audit or review and feedback of provider prescribing or AMU practices (n = 7 studies, 46.7%) and education (n = 6, 40%) were the most common intervention types implemented. Four studies assessed the impact of introducing various online or application-based diagnostic and clinical support tools, three studies introduced prescribing restrictions or pre-authorization requirements, two studies involved the development and/or introduction of a clinical treatment guideline, and one study implemented a diagnostic stewardship strategy to withhold laboratory results if there were no white blood cells or bacteria identified on microscopy.

Six of 42 of the studies (14.3%) that reported on the proportion of patients who received an antimicrobial documented a statistically significant (p value <0.05) increase in AMU after the

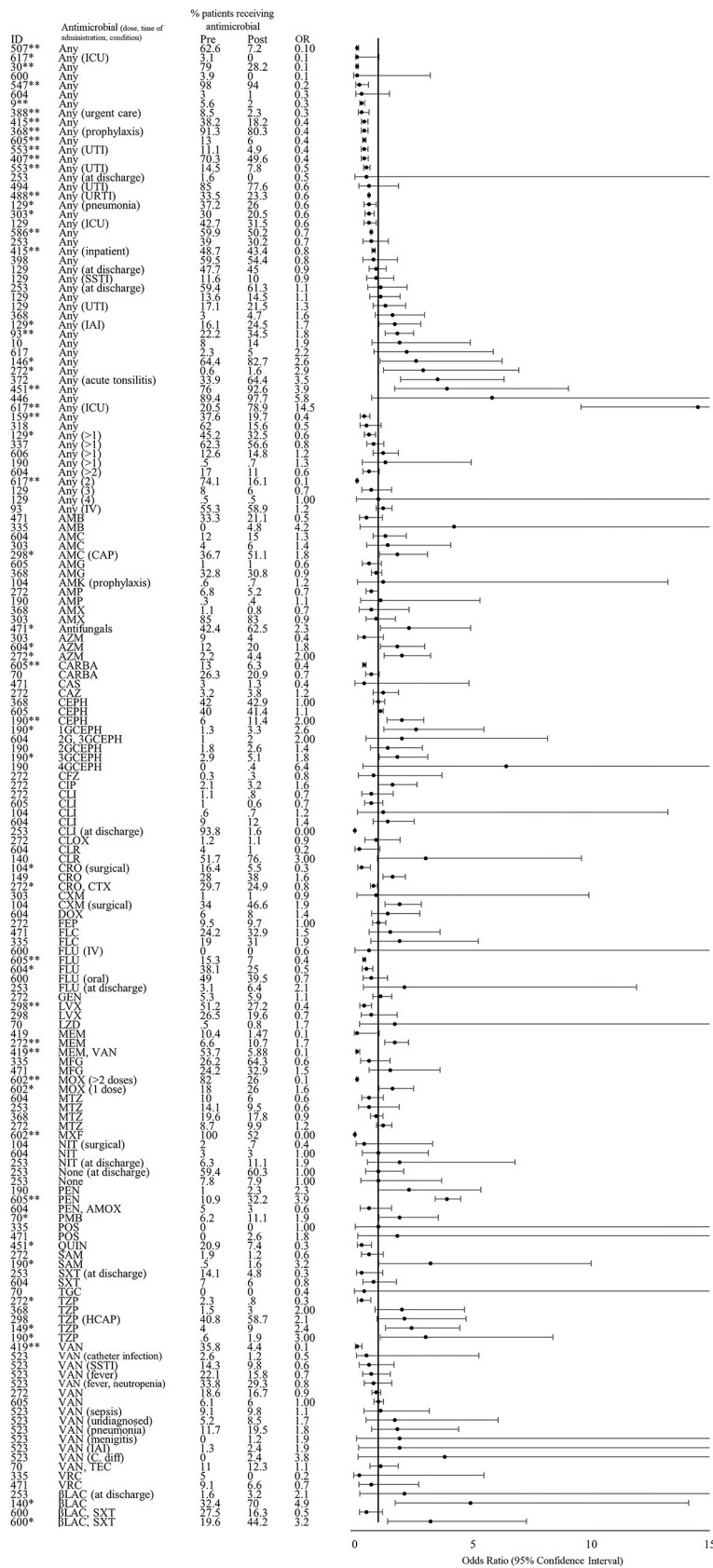

**Fig 3. Effect of interventions on antimicrobial use (proportion of patients receiving antimicrobials).** *Indicates significance <0.05. **Indicates significance <0.001. Antimicrobial abbreviations: AMK- amikacin, AMX- amoxicillin, AMC- amoxicillin-clavulanic acid, AMP- ampicillin, CARB- carbapenems, CTX- cefotaxime, CIP- ciprofloxacin, FOF- Fosfomycin, IPM- imipenem, LVX- levofloxacin, MDR- multi-drug resistant, MEM- meropenem, MET- methicillin, NIT- nitrofurantoin, TZP- piperacillin-tazobactam, SXT- trimethoprim-sulfamethoxazole, 3G-CEPH- third-generation cephalosporins. Other abbreviations: HCAP- healthcare associated pneumonia, IV- intravenous, CAP- community-acquired pneumonia, ICU- intensive care unit, IAI- intraabdominal infection, UTI- urinary tract infection, SSTI- skin or soft tissue infection.

intervention; the percentage increase averaged 10.01% (SD: 14.5) and ranged from 1.1 to 58.4%. All six of these studies described interventions in high-income countries (Denmark, Japan, Australia, USA, Spain, and Ireland); clinical settings of interventions included secondary or tertiary hospitals (n = 3 studies), academic or university hospitals (n = 2), and a children's hospital. The majority of these 6 studies implemented bundled interventions (n = 5, 83.3%); 4 included an audit or review and feedback activity; 3 involved the development and/or implementation of a clinical treatment guideline, protocol, or policy; 2 involved AMS education or training; 1 study introduced an AMS application; and another study introduced a rapid blood culture diagnostic test. Intervention duration ranged from 2 months to 45 months.

Seven studies (16.7%) reported simultaneous increases and decreases in different antimicrobial agents and classes that were statistically significant, ranging from a 21.3% pre- to post-intervention increase to a 56% decrease. For example, Leis *et al.*, reported a significant reduction in the proportion of patients prescribed fluoroquinolone (15.3 to 7%, p<0.001) and carbapenems (13 to 6.3%, p<0.001) and an increase in the proportion of patients prescribed penicillin (10.9 to 32.2%, p<0.001) after an intervention that implemented a point-of-care β-lactam allergy skin test among other AMS activities [41]. Another intervention described by Yogo *et al.* aimed to reduce broad-spectrum antibiotic prescribing rates and treatment durations at time of hospital discharge by introducing institutional guidance for oral step-down antibiotic selection and duration of therapy and implemented a pharmacy audit and real-time recommendations of discharge prescriptions [42]. This study documented a significant pre- to post-intervention decrease in fluoroquinolone use (38.1 to 25%, p = 0.001) and a significant increase in the proportion of patients prescribed azithromycin (12 to 20%, p = 0.01). Across all datapoints reported in these 7 studies, there was an average 1.8% decrease in the proportion of patients receiving antimicrobials.

**Theme 2: Appropriateness of therapy and adherence to treatment guidelines, protocols, and policies.** We extracted data from 38 studies that described pre- to post-intervention changes in the adherence/concordance of AMU or prescribing practices with local, national, or international treatment guidelines or protocols (Fig 4). Of those, 28 studies (73.7%) reported a significant improvement in appropriateness and/or adherence while 1 study (2.6%) reported a significant decline. Most studies that reported improvements in appropriateness of therapy or adherence to guidelines were conducted in high-income countries (n = 20, 71.4%, Australia; Canada, n = 2; Denmark; Germany; Greece; Ireland; Israel; Italy, n = 3); Netherlands; USA, n = 8) with 2 studies in upper-middle-income (Malaysia, South Africa), 5 in lower-middle (Egypt; India, n = 2; Kenya; Pakistan), and 1 in low-income countries (Nepal). Most studies assessed bundled interventions (n = 19, 67.9%) in secondary, tertiary, or academic/teaching hospital settings. Audit or review and feedback; the development of implementation of clinical treatment guidelines, protocols, or policies; and education or training were the most common interventions implemented as the sole intervention or in combination with other intervention activities in 18 (64.3%), 16 (57.4%), and 14 (50%) studies, respectively. Intervention duration ranged from 3 months to 6 years. The average increase in the proportion

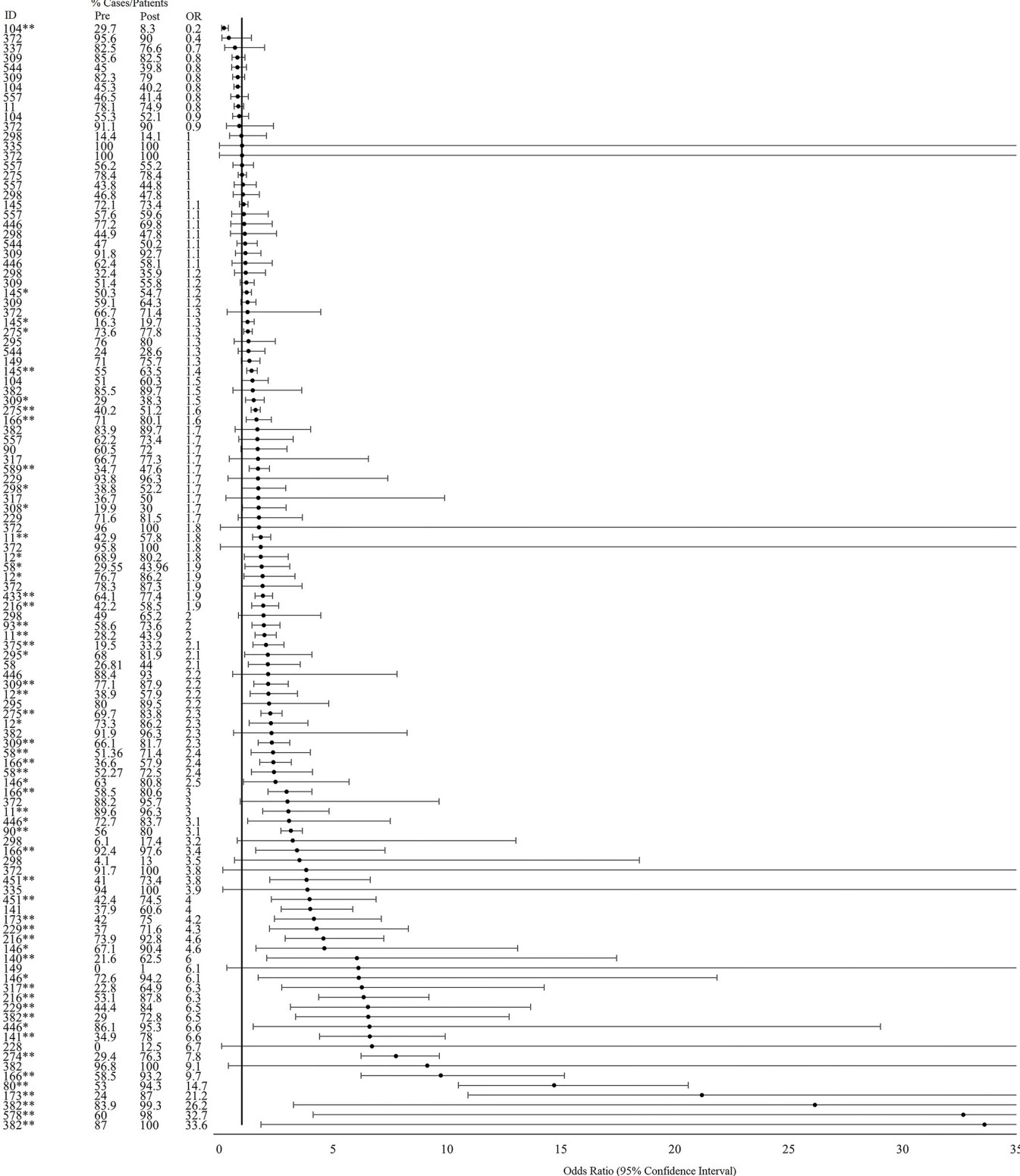

**Fig 4. Effect of interventions on appropriateness of antimicrobial therapy or adherence to clinical guidelines (including drug selection, dosage, duration, and timing of administration).** *Indicates significance <0.05. **Indicates significance <0.001.

of cases or patients treated with appropriate or guideline-concordant antimicrobial therapies across all studies and indicators where a statistically significant change was reported was 20.8% (SD: 12.3). The study that reported a decline in guideline concordance was conducted in a secondary care hospital in the United Arab Emirates; this study assessed the impact of a 3-month intervention to develop and implement a new treatment guideline and conduct educational lectures for physicians for adopting the guidelines into their practice. The proportion of patients who received antimicrobial treatment that complied with guidelines regarding the timing of discontinuation fell from 29.7 to 8.3% (Odds ratio: 0.2, 95% CI: 0.1–0.4; Chi square p<0.001).

**Theme 3: Antimicrobial stewardship.** We extracted AMS data from 20 studies; 9 (45%) of which demonstrated significant improvements in AMS practices and 1 (5%) that reported a significant reduction in AMS practices (Fig 5). Two of three studies (66.7%) that assessed changes in intravenous to oral stepdown reported pre- to post-intervention improvements in conversion rates while 7 of 18 studies (38.8%) that assessed changes in the proportion of patients or cases where a culture or clinical diagnostic test was obtained reported significant pre- to post- increases. Of the studies the reported significant improvements in AMS, 3 were conducted in lower-middle-income countries (Egypt, India, Vietnam), 2 in upper-middle-income countries (China, South Africa), and 4 in high-income countries (Canada; USA,

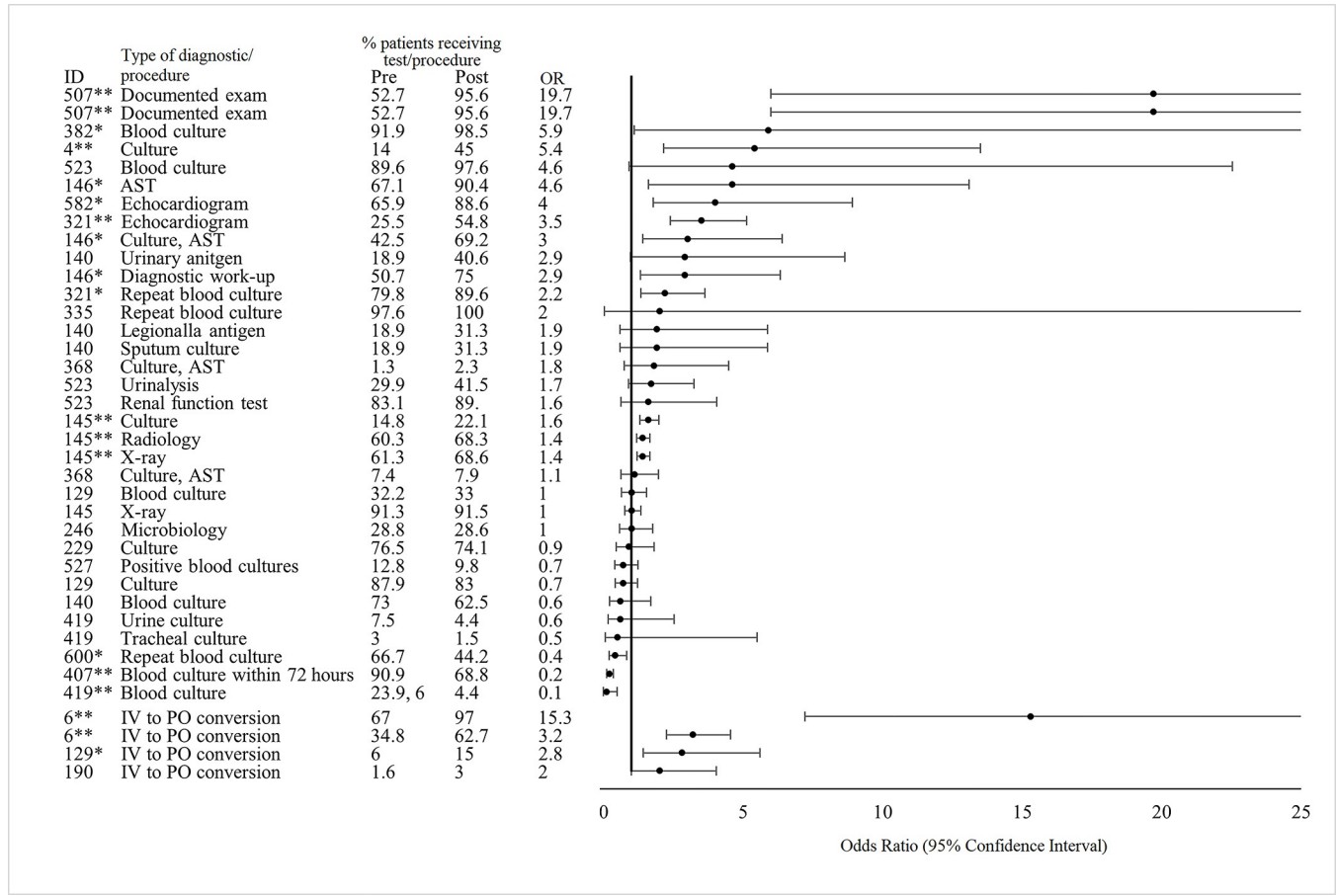

**Fig 5. Effect of interventions on the utilization of clinical diagnostic and laboratory tests/cultures and intravenous to oral conversion rates.** *Indicates significance <0.05. **Indicates significance <0.001. Abbreviations: AST- antimicrobial susceptibility testing, IV- intravenous, PO- per os (by mouth/oral).

n = 4). Most studies assessed bundled interventions (n = 6, 66.7%) that were implemented in tertiary or academic health facilities (n = 6, 66.7%). Four interventions involved the development and/or dissemination of treatment guidelines, protocols, or policies; 4 involved education or training sessions; 4 included the introduction of or changes to antibiotic order forms or clinical-support tools; 3 interventions involved audit or review and feedback; 1 study included the development of a facility antibiogram. Intervention durations ranged from 2 months to 24 months. One study assessed the impact of a one-year audit/review and feedback intervention on prescribers at a pediatric hospital in Iran obtaining blood cultures to aid in diagnosis and found that the proportion of patients whose care included a blood culture for diagnosis declined from 23.9% pre-intervention to 4.4% post-intervention (OR: 0.1, 95% CI: 0.04–0.5; Chi square p<0.001). The study also reported non-significant pre- to post-intervention declines in the proportion of patients whose diagnosis was aided by tracheal and urine cultures.

**Theme 4: Antimicrobial resistance.** Of the studies that met our inclusion criteria and reported on our primary outcomes of interest, 11 reported on our secondary outcome of pre- to post-intervention changes in AMR (Fig 6). Across these studies, data was reported for 60 unique organism-antimicrobial combinations. Five studies (45.5%) reported significant decreases in the proportion of isolates that were resistant to antibiotics or the proportion of patients with drug-resistant infections across 17 antimicrobial-organism combinations while 3 studies (27.3%) reported both significant increases and decreases in resistance rates for different antimicrobial-organism combinations. All 8 studies reporting AMR data were conducted in high-income countries (Canada, Israel, France, Japan, Korea, Spain, and the USA; n = 2). The 5 studies that reported only decreases in resistance rates were conducted in either acute care hospitals, nursing homes, public hospitals, or teaching hospitals (n = 2); most of the studies tested single activity interventions (n = 4, 80%) including introducing a treatment management protocol, and audit or review and feedback (n = 3). The bundled intervention consisted of the implementation of an AMS toolkit and subsequent education and an antibiotic mobile team for daily AMS coordination.

The 3 studies that reported a combination of significant pre- to post-intervention changes were conducted in nursing homes or tertiary/teaching hospitals (n = 2); two of these interventions consisted of bundled AMS activities. One intervention described by Ziv-On *et al.* consisted of review and feedback, education, and restricted use of certain antibiotics that required pre-prescription authorization [43]. There were significant pre- to post-intervention decreases in the proportion of cultured isolates resistant to gentamicin (19% to 15.4%, OR: 0.8, 95% CI: 0.677–0.888, Chi-square p<0.001) and ciprofloxacin (33.8% to 29.9%, OR: 0.8, 95% CI: 0.7–0.9, Chi-square p = 0.001) and significant increases in the proportion of isolates resistant to cefuroxime (26.5 to 33.8%, OR: 1.4, Chi-square p<0.001) and amoxicillin-clavulanic acid (29.7 to 32.6%, OR: 1.1, 95% CI: 1.03–1.3, Chi-square p = 0.007). There were decreases in average quarterly use of aminoglycoside (gentamicin) from 2.3 (SD: 0.2) to 1.7 (SD: 0.3) daily defined doses/100 days of admission, quinolones (ciprofloxacin) 10.9 (SD: 0.7) to 6.71 (SD: 1.8), second-generation cephalosporins (cefuroxime) from 11.3 (SD: 0.7) to 8.6 (SD: 2.7), and amoxicillin-clavulanic acid from 19.8 (SD: 4.7) to 9.9 (SD: 3.7).

The second bundled study by Tandan *et al.* consisted of the introduction of a treatment guideline, education, and audit and feedback [44]. There were significant decreases in the proportion of cultured *E. coli* isolates resistant to nitrofurantoin (13.2 to 6.6%, OR: 0.5, 95% CI: 0.3–0.7. Chi-square p<0.001) and trimethoprim-sulfamethoxazole (40.4 to 27.8%, OR: 0.6, 95% CI: 0.4–0.7, Chi-square p<0.001) and *Klebsiella pneumoniae* and Enterobacteriaceae to nitrofurantoin (64.4 to 47.8, OR: 0.5, 95% CI: 0.334–0.768; 43.9 to 38.8%, OR: 0.8, 95% CI: 0.7–1.0, respectively) but a significant increase in the proportion of *Proteus* spp. isolates that

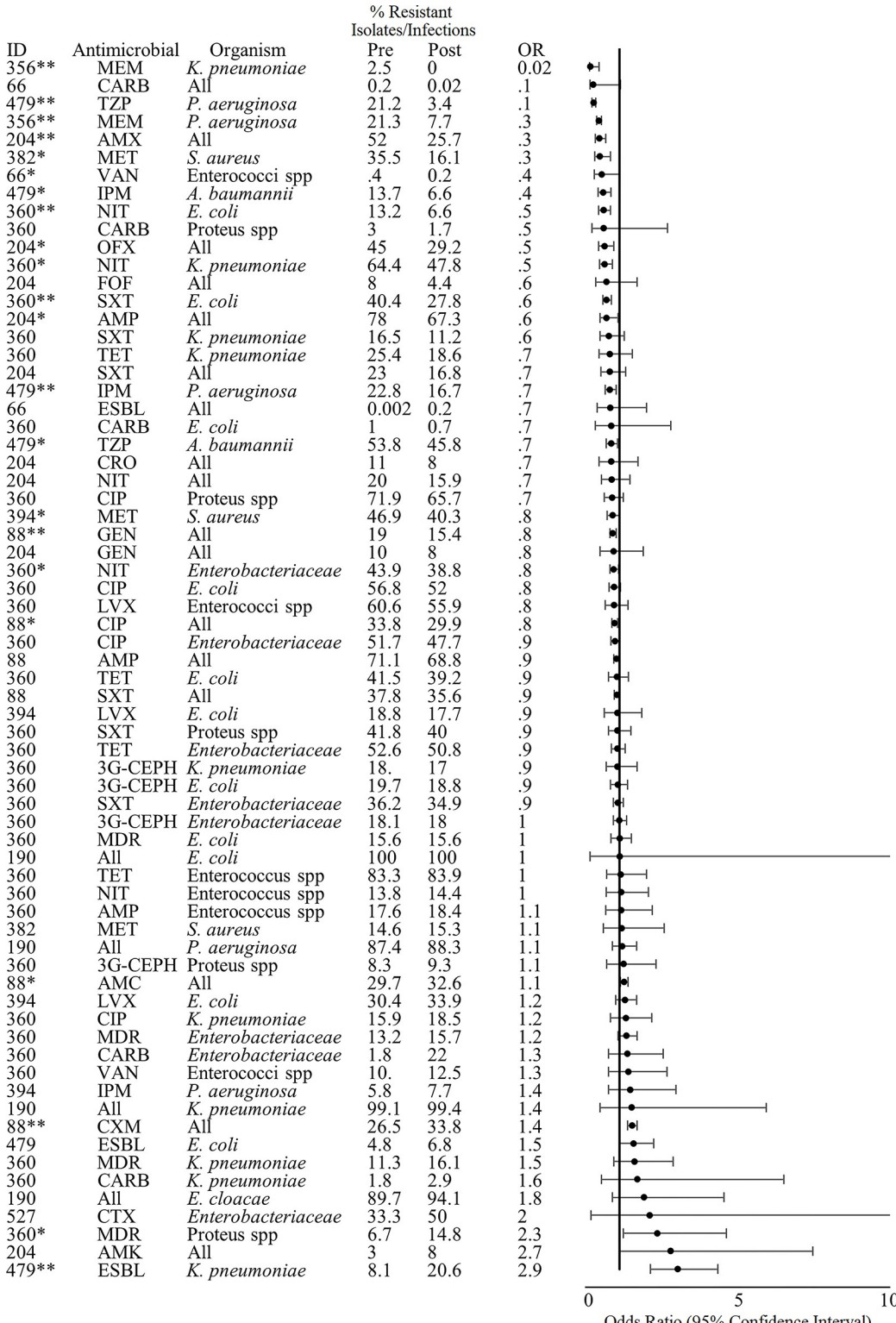

| ID | Antimicrobial | Organism | % Resistant Isolates/Infections Pre | % Resistant Isolates/Infections Post | OR |
|---|---|---|---|---|---|
| 356** | MEM | *K. pneumoniae* | 2.5 | 0 | 0.02 |
| 66 | CARB | All | 0.2 | 0.02 | .1 |
| 479** | TZP | *P. aeruginosa* | 21.2 | 3.4 | .1 |
| 356** | MEM | *P. aeruginosa* | 21.3 | 7.7 | .3 |
| 204** | AMX | All | 52 | 25.7 | .3 |
| 382* | MET | *S. aureus* | 35.5 | 16.1 | .3 |
| 66* | VAN | Enterococci spp | .4 | 0.2 | .4 |
| 479* | IPM | *A. baumannii* | 13.7 | 6.6 | .4 |
| 360** | NIT | *E. coli* | 13.2 | 6.6 | .5 |
| 360 | CARB | Proteus spp | 3 | 1.7 | .5 |
| 204* | OFX | All | 45 | 29.2 | .5 |
| 360* | NIT | *K. pneumoniae* | 64.4 | 47.8 | .5 |
| 204 | FOF | All | 8 | 4.4 | .6 |
| 360** | SXT | *E. coli* | 40.4 | 27.8 | .6 |
| 204* | AMP | All | 78 | 67.3 | .6 |
| 360 | SXT | *K. pneumoniae* | 16.5 | 11.2 | .6 |
| 360 | TET | *K. pneumoniae* | 25.4 | 18.6 | .7 |
| 204 | SXT | All | 23 | 16.8 | .7 |
| 479** | IPM | *P. aeruginosa* | 22.8 | 16.7 | .7 |
| 66 | ESBL | All | 0.002 | 0.2 | .7 |
| 360 | CARB | *E. coli* | 1 | 0.7 | .7 |
| 479* | TZP | *A. baumannii* | 53.8 | 45.8 | .7 |
| 204 | CRO | All | 11 | 8 | .7 |
| 204 | NIT | All | 20 | 15.9 | .7 |
| 360 | CIP | Proteus spp | 71.9 | 65.7 | .7 |
| 394* | MET | *S. aureus* | 46.9 | 40.3 | .8 |
| 88** | GEN | All | 19 | 15.4 | .8 |
| 204 | GEN | All | 10 | 8 | .8 |
| 360* | NIT | *Enterobacteriaceae* | 43.9 | 38.8 | .8 |
| 360 | CIP | *E. coli* | 56.8 | 52 | .8 |
| 360 | LVX | Enterococci spp | 60.6 | 55.9 | .8 |
| 88* | CIP | All | 33.8 | 29.9 | .8 |
| 360 | CIP | *Enterobacteriaceae* | 51.7 | 47.7 | .9 |
| 88 | AMP | All | 71.1 | 68.8 | .9 |
| 360 | TET | *E. coli* | 41.5 | 39.2 | .9 |
| 88 | SXT | All | 37.8 | 35.6 | .9 |
| 394 | LVX | *E. coli* | 18.8 | 17.7 | .9 |
| 360 | SXT | Proteus spp | 41.8 | 40 | .9 |
| 360 | TET | *Enterobacteriaceae* | 52.6 | 50.8 | .9 |
| 360 | 3G-CEPH | *K. pneumoniae* | 18. | 17 | .9 |
| 360 | 3G-CEPH | *E. coli* | 19.7 | 18.8 | .9 |
| 360 | SXT | *Enterobacteriaceae* | 36.2 | 34.9 | .9 |
| 360 | 3G-CEPH | *Enterobacteriaceae* | 18.1 | 18 | 1 |
| 360 | MDR | *E. coli* | 15.6 | 15.6 | 1 |
| 190 | All | *E. coli* | 100 | 100 | 1 |
| 360 | TET | Enterococcus spp | 83.3 | 83.9 | 1 |
| 360 | NIT | Enterococcus spp | 13.8 | 14.4 | 1 |
| 360 | AMP | Enterococcus spp | 17.6 | 18.4 | 1.1 |
| 382 | MET | *S. aureus* | 14.6 | 15.3 | 1.1 |
| 190 | All | *P. aeruginosa* | 87.4 | 88.3 | 1.1 |
| 360 | 3G-CEPH | Proteus spp | 8.3 | 9.3 | 1.1 |
| 88* | AMC | All | 29.7 | 32.6 | 1.1 |
| 394 | LVX | *E. coli* | 30.4 | 33.9 | 1.2 |
| 360 | CIP | *K. pneumoniae* | 15.9 | 18.5 | 1.2 |
| 360 | MDR | *Enterobacteriaceae* | 13.2 | 15.7 | 1.2 |
| 360 | CARB | *Enterobacteriaceae* | 1.8 | 22 | 1.3 |
| 360 | VAN | Enterococci spp | 10. | 12.5 | 1.3 |
| 394 | IPM | *P. aeruginosa* | 5.8 | 7.7 | 1.4 |
| 190 | All | *K. pneumoniae* | 99.1 | 99.4 | 1.4 |
| 88** | CXM | All | 26.5 | 33.8 | 1.4 |
| 479 | ESBL | *E. coli* | 4.8 | 6.8 | 1.5 |
| 360 | MDR | *K. pneumoniae* | 11.3 | 16.1 | 1.5 |
| 360 | CARB | *K. pneumoniae* | 1.8 | 2.9 | 1.6 |
| 190 | All | *E. cloacae* | 89.7 | 94.1 | 1.8 |
| 527 | CTX | *Enterobacteriaceae* | 33.3 | 50 | 2 |
| 360* | MDR | Proteus spp | 6.7 | 14.8 | 2.3 |
| 204 | AMK | All | 3 | 8 | 2.7 |
| 479** | ESBL | *K. pneumoniae* | 8.1 | 20.6 | 2.9 |

Odds Ratio (95% Confidence Interval) — 0, 5, 10

**Fig 6. Effect of interventions on the proportion of isolates/infections resistant to various antimicrobials.** *Indicates significance <0.05. **Indicates significance <0.001. Antimicrobial Abbreviations: AMK- amikacin, AMX- amoxicillin, AMC- amoxicillin-clavulanic acid, AMP- ampicillin, CARB- carbapenems, CTX- cefotaxime, CIP- ciprofloxacin, FOF- Fosfomycin, IPM- imipenem, LVX- levofloxacin, MDR- multi-drug resistant, MEM- meropenem, MET- methicillin, NIT- nitrofurantoin, TZP- piperacillin-tazobactam, SXT- trimethoprim-sulfamethoxazole, 3G-CEPH- third-generation cephalosporins.

were multi-drug resistant (6.7 to 14.8%, OR: 2.3, 95% CI: 1.1–4.6, Chi-square p = 0.02). The study did not report data on changes in AMU.

The third intervention described by Kim *et al.* tested the impact of restricting prescription of third-generation cephalosporin through a computerized antimicrobial prescription program [45]. There were significant decreases in the proportion of *P. aeruginosa* isolates resistant to imipenem (22.8% to 16.7%, OR: 0.6, 95% CI: 0.5–0.9) and *A. baumannii* isolates resistant to imipenem (13.7% to 6.6%, OR: 0.4, 95% CI: 0.3–0.7) and trimethoprim-sulfamethoxazole (53.8% to 45.8%, OR: 0.7, 95% CI: 0.6–0.9) but a significant increase in the proportion of extended-spectrum beta-lactamase (ESBL)-producing *K. pneumoniae* isolates (8.1% to 20.6%, OR: 2.9, 95% CI: 2.0–4.3). There was no significant change in carbapenem (imipenem) or beta-lactamase inhibitor use, and data on sulfonamide (trimethoprim-sulfamethoxazole) use was not reported.

**Theme 5: Clinical outcomes.** We extracted clinical outcome data from 57 of 301 total studies (Fig 7). Of three studies that reported the proportion of patients who were admitted to the intensive care unit before and after the intervention, only one reported a statistically significant change, a decrease from 44.5% to 35.5% of patients (OR: 0.7, 95% CI: 0.5–1.0) [46]. Three separate studies reported the proportion of patients who experienced an adverse event before and after the intervention, and one reported a statistically significant change, an increase from 1.1% to 10.0% (OR: 9.9, 95% CI: 8.4–11.6) [47]. Of 41 total studies that reported data on patient mortality before and after AMS interventions, 6 (14.6%) reported statistically significant decreases in the proportion of patients who died while 2 (4.8%) reported statistically significant increases in the proportion of patients who died; all other studies reported no significant change in mortality rates. Of 18 studies that reported on readmission rates, one study reported a significant increase and one a significant decrease in readmission after the intervention compared to before. Six of 8 studies (75%) that recorded the proportion of patients who experienced a recurrence or re-infection reported no significant pre- to post-intervention while two (25%) reported significant decreases. Finally, three studies reported on the proportion of patients experiencing treatment failure before and after interventions with only one reporting a significant pre- to post-intervention change, a decrease from 33.3% to 14.0% (OR: 0.3, 95% CI: 0.2–0.5) [48].

Based on univariable t-tests and multivariable linear regression, there were no statistically significant (p<0.05) associations between pre- to post-intervention absolute differences and intervention type, duration, or setting of intervention across any of our study outcomes in Themes 1–5.

## Discussion

Overall, there was a paucity of studies that assessed the impact of behavior-change interventions on AMU, AMS, and AMR in the animal health sector; further evidence is needed to understand best practices in these settings. A minority of studies conducted in the human health sector (35.7%) reported significant (p<0.05) pre- to post-intervention decreases in AMU while a majority (73.7%) reported significant improvements in adherence of antimicrobial therapies to local, national, or international guidelines. Of 20 studies, 45% demonstrated significant improvements in AMS practices while only 1 study reported a significant reduction in AMS practices. Eleven studies reported on pre- to post-intervention changes in the AMR burden, all of which took place in high-income countries. Five of those studies (45.5%) reported significant decreases in the proportion of isolates that were resistant to antibiotics or the proportion of patients with drug-resistant infections across 17 antimicrobial-organism combinations; 3 studies (27.3%) reported both significant increases and decreases in resistance

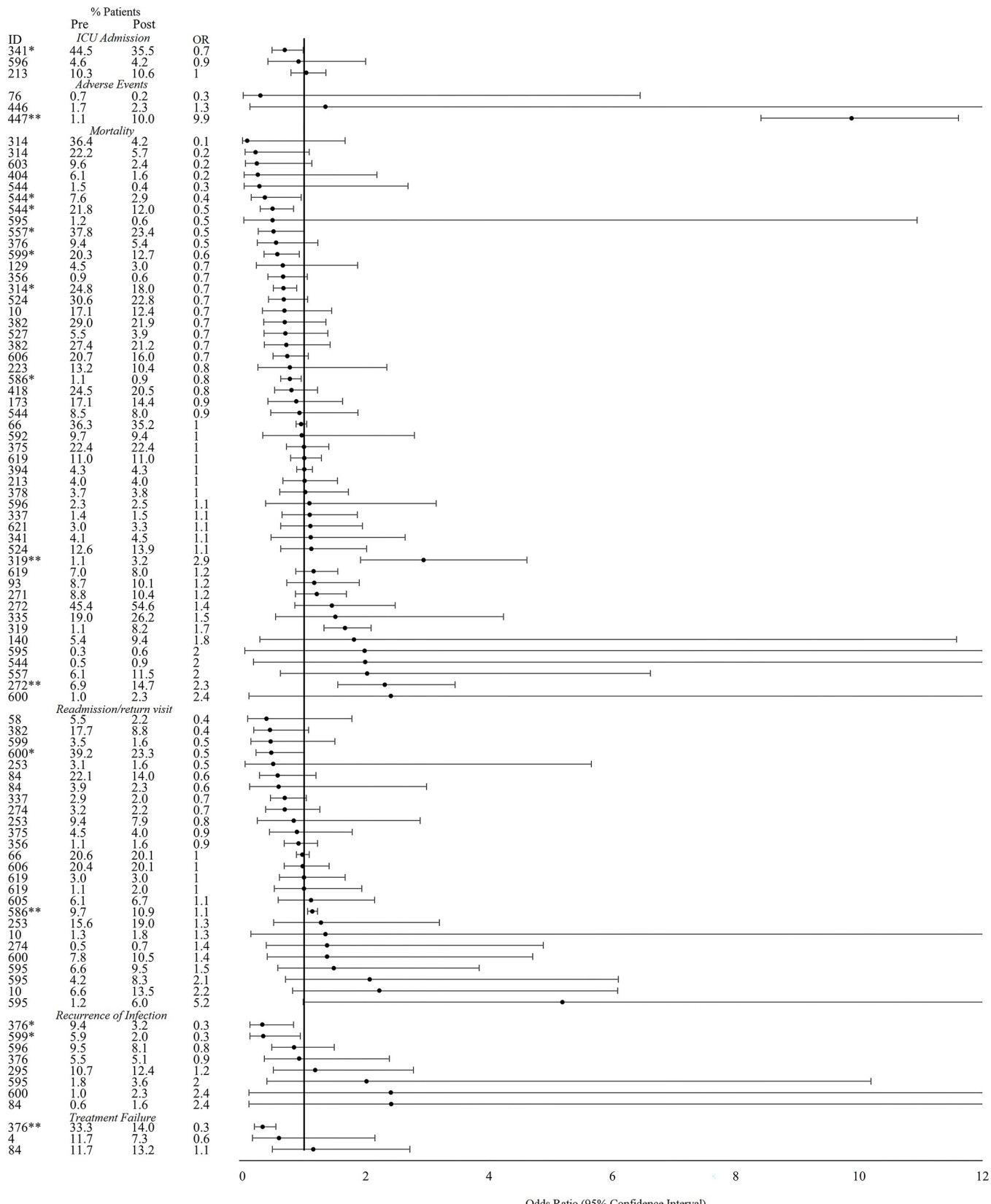

**Fig 7. Effect of interventions on clinical outcomes (ICU admission rate, adverse events, mortality, readmission or return visit to a health facility, recurrence of infection, and treatment failure).** *Indicates significance <0.05. **Indicates significance <0.001. Abbreviations: ICU- intensive care unit.

rates for different antimicrobial-organism combinations. Generally, significant pre- to post-intervention changes in clinical outcomes–including ICU admission, mortality, adverse events, reinfection, mortality, and treatment failure rates–were infrequent with studies more commonly reporting improvements across those outcomes. Our study found no intervention characteristics that were associated with overall success the various thematic areas in any setting or subsetting included.

This study had several limitations. Unfortunately, the small number of animal health studies that met this review's inclusion criteria and variation across the type of statistic reported, the study design, and the outcome variables reported by studies pertaining to the human health sector precluded meta-analyses, and we were only able to present descriptive, summary analyses. In addition, we limited our review to peer-reviewed studies which may have overlooked a significant number of ongoing interventions in various settings to improve AMR, AMU, and AMS and may have also introduced a publication bias where successful studies are more likely to be more published than unsuccessful interventions. Moreover, we only considered interventions that targeted behavior change among stakeholder groups involved in the use or prescription of antimicrobials thereby overlooking other important interventions such as those that aim to increase awareness and knowledge of AMR, AMU, and AMS; those that effect funding levels or policies; those that change access to clinically appropriate antimicrobials or other critical diagnostic and laboratory testing infrastructure; among other interventions and efforts that can effectively improve AMU and AMS and reduce AMR amongst human and animal populations. There were also limitations in the studies included in our final analysis. First, the majority of studies took place in developed countries; given the disparities in healthcare infrastructure, human resources, and AMR burden in high- versus low-resource settings, our findings may not be globally applicable. Notably, adherence to guidelines was the most improved intervention outcome in the human health sector; however, in many LMICs, local or national clinical treatment guidelines are often unavailable or outdated. A 2021 review of antimicrobial treatment guidelines in Africa found that only 20 of the 55 Africa Union member states had national guidelines at the time of the study, and only 10 member states had updated those guidelines since 2015 [49].

There were also limitations in the quality of evidence provided by the included studies. Few interventions utilized random allocation of treatment groups; most interventions were assessed using pre/post observational studies with no control group. The duration of interventions varied widely with many only assessing short-term impacts of interventions. Moreover, few studies described the theories underpinning behavior-change interventions which may improve effectiveness [50]. Our quality assessment examined the risk of selection bias, study design, data collection methods, intervention integrity, and analyses, among other criteria. Overall, 4 of 11 studies (36.4%) that described interventions in the animal health or agriculture sectors and 129 of the 290 studies (44.5%) that described interventions in the human health sector were determined to have strong overall quality; therefore, there is room for improvement in overall study quality.

Overall, there is a need to increase the evidence base of behavior-change interventions regarding AMU, AMS, and AMR in the animal health and LMIC settings. Due to variation in the study settings and designs, it is difficult to identify which practices are most impactful across the board. There is a need to systematize approaches to assessing the impact of interventions and for studies to collect and report data representing the multiple ways AMR and AMS impacts clinical practice and outcomes so that comparative analysis can reveal best practices in various settings and those practices, interventions, or stewardship approaches can be scaled locally and globally. However, given the diverse span of settings in which antimicrobials are used and variation in stewardship interventions and approaches, it may be necessary to

develop more precise metrics to assess or monitor intervention impact in more specific or sensitive ways than the approach taken in this study. Regardless of study setting, we encourage future studies to consider the complexity of the drivers and impacts of AMU and AMR and the various ways in which AMS efforts can impact the patient and provider behavior, the quality and cost of care, and patient outcomes, in addition to AMU and AMR rates. For example, studies that aim to assess the impacts of AMS should not only measure overall AMU rates but the clinical appropriateness of therapy; adherence to local, national, or international guidelines; clinical outcomes, AMS practices, cost and duration of patient stays; and clinical outcomes. Identifying key metrics of stewardship success and tailoring those metrics to various settings represents a potential next step to advance stewardship research and policy efforts across the diverse settings where antimicrobials are used.

## Supporting information

**S1 Checklist. PRISMA checklist.**
(DOCX)

**S1 File. Title and publication information for all animal health studies that met inclusion criteria.**
(XLSX)

**S2 File. Title and publication information for all human health studies that met inclusion criteria.**
(XLSX)

**S3 File. Final quality assessment scores for all studies that met the inclusion criteria.**
(XLSX)

**S4 File. Database indexing study information, intervention description, and reported and calculated statistics for all studies included in analyses.**
(XLSX)

## Acknowledgments

We thank Nathan Freeman for assistance with the quality assessment.

## Author Contributions

**Conceptualization:** Jessica Craig, Aditi Sriram, Wendy Beauvais.

**Data curation:** Jessica Craig, Aditi Sriram, Rachel Sadoff, Sarah Bennett, Felix Bahati.

**Formal analysis:** Jessica Craig, Rachel Sadoff, Sarah Bennett, Felix Bahati.

**Investigation:** Jessica Craig, Aditi Sriram.

**Methodology:** Jessica Craig, Aditi Sriram, Wendy Beauvais.

**Project administration:** Jessica Craig, Aditi Sriram, Wendy Beauvais.

**Software:** Jessica Craig.

**Supervision:** Jessica Craig, Wendy Beauvais.

**Validation:** Jessica Craig, Aditi Sriram.

**Visualization:** Jessica Craig, Wendy Beauvais.

**Writing – original draft:** Jessica Craig.

**Writing – review & editing:** Rachel Sadoff, Felix Bahati, Wendy Beauvais.

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
