## [Decision Letter · Decision Letter 0]

27 Mar 2023

PGPH-D-22-02066

Behavior-change interventions to improve antimicrobial stewardship in human health, animal health, and livestock agriculture: A systematic review

Dear Ms. Craig,

Thank you for submitting your manuscript to PLOS Global Public Health. After careful consideration, we feel that it has merit but does not fully meet PLOS Global Public Health’s publication criteria as it currently stands. Therefore, we invite you to submit a revised version of the manuscript that addresses the points raised during the review process.

A rebuttal letter that responds to each point raised by the reviewers. You should upload this letter as a separate file labeled 'Response to Reviewers'.A marked-up copy of your manuscript that highlights changes made to the original version. You should upload this as a separate file labeled 'Revised Manuscript with Track Changes'.An unmarked version of your revised paper without tracked changes. You should upload this as a separate file labeled 'Manuscript'.

We look forward to receiving your revised manuscript.

Kind regards,

Sumanth Gandra

Academic Editor

Journal Requirements:

Reviewers' comments:

Reviewer's Responses to Questions

**Comments to the Author**

1. Does this manuscript meet PLOS Global Public Health’s publication criteria? Is the manuscript technically sound, and do the data support the conclusions? The manuscript must describe methodologically and ethically rigorous research with conclusions that are appropriately drawn based on the data presented.

Reviewer #1: Yes

Reviewer #2: Yes

2. Has the statistical analysis been performed appropriately and rigorously?

Reviewer #1: Yes

Reviewer #2: Yes

3. Have the authors made all data underlying the findings in their manuscript fully available (please refer to the Data Availability Statement at the start of the manuscript PDF file)?

Reviewer #1: Yes

Reviewer #2: Yes

4. Is the manuscript presented in an intelligible fashion and written in standard English?

Reviewer #1: Yes

Reviewer #2: Yes

5. Review Comments to the Author

Reviewer #1: The manuscript titled: "Behavior-change interventions to improve antimicrobial stewardship in human health, animal health, and livestock agriculture: A systematic review", is one that has contributed greatly to knowledge but requires minor corrections.

Specific comment:

L127: What is 'risk ratio'? What is it? It is only "Odd ratio' or relative risk'. Delete it.

L148: Delete 'rate ratio'.

Also, check and correct similar wrongly used statistical phrases.

Reviewer #2: In this paper, Craig and co-authors attempt to summarize the current state of knowledge regarding the efficacy of behavior-change interventions relevant to antimicrobial stewardship on five domains of outcomes – stewardship practices, antibiotic use, antibiotic resistance, guideline adherence, and clinical outcomes. They were able to identify 301 mostly moderate or low quality studies that they analyzed under each domain. For the studies relating to antibiotic use in animals, there were too few to do a detailed statistical analysis. Unsurprisingly, their findings do not illuminate whether certain interventions are likely to be more successful than others – the heterogeneity of intervention types, locations, patient populations, and clinical contexts almost guarantees that no overarching conclusions can be drawn. This is a common theme in stewardship literature as antibiotic use and interventions to influence prescriber behavior are highly context dependent and almost always difficult to generalize. Still, overall this is a useful paper that acts as a survey of the field at one point in time and the finding that essentially all studies reported either no effect on, or improvement of typical safety/quality outcomes such as mortality and infection relapse rates is a valuable one. The paper is broad in its scope, which could be a strength or a weakness, but I appreciated the authors’ attempt to comprehensively report out their findings in a single paper. The paper is well written and requires only minor edits.

Background: This section can be shortened. It is non-controversial at this point that non-pharmaceutical/behavioral interventions to reduce antibiotic use are increasingly necessary. In line 67, the authors write that in the past 50 years no new antimicrobial agents active against Gram-negatives have been introduced in the last 50 years. This is simply not true – looking back at the document they cite, it seems that they meant to say that no new _classes_ of antibacterial active against Gram-negatives have been developed, which is true. The word “Gram” should be capitalized.

Methods: No specific changes. The decision to compare AMU data as the proportion of patients receiving antibiotics as opposed to more typical metrics used in stewardship papers like DDDs or antimicrobial days per days present I think is justifiable.

Results: No specific changes other than “gentamicin” is spelled incorrectly throughout.

Discussion: The authors conclude that that there is a need to systematize approaches for assessing the impact of interventions by developing more effective metrics. This is reasonable and echoes the popular sentiment, but I think that another conclusion that could be drawn from this review is that given the sheer diversity of contexts in which antibiotics are used, the diversity of interventions, and the heterogeneity of implementations of the same intervention, that perhaps this is an impossible aspiration. And if that’s the case, the question becomes what would be an alternative way to approach the development and reporting of stewardship interventions. If the authors trim some of their introduction, it might be worthwhile to expand on that idea in the discussion, but this is optional.

6. PLOS authors have the option to publish the peer review history of their article (what does this mean?). If published, this will include your full peer review and any attached files.

**Do you want your identity to be public for this peer review?** For information about this choice, including consent withdrawal, please see our Privacy Policy.

Reviewer #1: No

Reviewer #2: No

---

## [Editor Report · Decision Letter 1]

18 Apr 2023

Behavior-change interventions to improve antimicrobial stewardship in human health, animal health, and livestock agriculture: A systematic review

PGPH-D-22-02066R1

Dear Ms. Craig,

We are pleased to inform you that your manuscript 'Behavior-change interventions to improve antimicrobial stewardship in human health, animal health, and livestock agriculture: A systematic review' has been provisionally accepted for publication in PLOS Global Public Health.

Best regards,

Sumanth Gandra

Academic Editor
